

# Study on connectivity mechanism and robustness of three - dimensional pore network of sandstone based on complex network theory

Guannan Liu [1,2] *,    Xiaopeng Pei [3] ,    Feng Gao [1,2] ,    Xin Liang [1,2] ,    Jianguo Wang [1,2] ,    Dayu Ye [3]

[1]Mechanics and Civil engineering Institute, China University of Mining and Technology, Xuzhou city, Jiangsu province 221000, China
[2] State Key Laboratory for Geomechanics and Deep Underground Engineering, China University of Mining and Technology, Xuzhou city, Jiangsu province 221000, China
[3] Chemical engineering Institute, China University of Mining and Technology, Xuzhou city, Jiangsu province 221000, China
* correspondence to Guannan Liu (guannanliu@cumt.edu.cn)

**Abstract:**    There are a large number of pores and throats inside the rock, with different magnitude and shape, whose connection is complex [1-3]. Based on the complex network theory, combined with X – ray CT scan and image processing technology, we used sandstone as an example to study the structural characteristics of rock network of different porosities. The experimental results show that the seepage network of sandstone is similar to the BA scale-free network in the structural characteristics. The average path length of sandstone *generally* increases with the increase of network magnitude. The average of number of edges of node plays a *dominant* role for the porosity of sandstone. It is inferred that in the large number of pores, few pores with a number of connections have an important role in the overall connectivity of the sandstone seepage network. At the same time, sandstone seepage network has better fault tolerance rate and *robustness* to external random attacks. The results of this paper may provide a new idea for the study of fluid storage and migration mechanisms in porous materials and the application of complex network theory in interdisciplinary fields.

**Key words:**  Complex network theory    sandstone    network topology    Seepage rate    robustness

## Introduction:
In recent years, with the increasing demand for oil and gas resources, the requirement of oil and gas exploration technology improves constantly, at the same time, the issues of water conservancy projects such as groundwater seepage, and the environment such as waste transfer and decomposition highlight constantly. Therefore, further understanding the rock pore structure, defining network structure characteristics, and revealing the microcosmic mechanism of macroscopic permeability of rock so as to grasp the geography and geologic characteristics more accurately has became a goal of scientists and engineers in numerous engineering fields.

However, achieving these goals faces numerous difficulties and challenges. As the distribution of rock pore is disorder, with large span scale and complex stress, and it is buried under the ground, it just likes





a "black box". All the time, most of the people study the mechanics and chemical properties of rock by experiment thus characterize rock pore and microscopic network indirectly. Due to the difficulty of modeling and calculation in theory, it is impossible to visualize intuitively and characterize quantitatively the internal structure of rock mass, and it is difficult to accurately grasp the inherent law

and essence of complex and changing physical and mechanical behavior of rock, therefore, we have to face the embarrassing situation that the physical and mechanical properties of the rock obtained by the limited sampling samples are discrete and not easy to apply and predict accurately. In fact, the network model has been studied for many years. The current research of it mainly focuses on the distribution and connection of pores and so on. In the wake of developments in science and technology, such as CT scan

technique, we can view the micro-state of rock on the micrometer scale and nano-scale [4-5]. Joshi [6] proposed the Gauss field algorithm that can be used to reconstruct porous media for the first time. Based on the statistical information of rock slice, porosity and two-point correlation function were used as the constraint condition; he obtained the model of the two-dimensional numerical core. Øren [7] reconstructed a sandstone structure model by the process method, and found that the model better

reflects the geometric structure and permeability of the real rock by comparing with the digital image of the core slice. Jun Yao [8] used the process method and simulated annealing algorithm to reconstruct the artificial core, sandstone and carbonate rock digital core. Xuetao Hu [9] used the directional seepage theory to establish a stochastic network model that can characterizes the micro-pore structure and wettability characteristics of the rock. Zhao [10] used the multidirectional scanning method to search the

pore space for different directions to obtain the pore structure model. Based on high-precision CT scan technique, Blunt [11] extracted the pore structure of different porous media, and introduced the research method of permeability of the pore scale. Dong [12] used the central axis method to extract and connect the central axis that connects the pores of the core, which can characterize the porosity of the rock accurately. However, the pores of the rock network are interconnected, and they are changing unceasing

under the action of the unavoidable external force so that the entire network changes irreversibly, which will cause great interference for study. Whether the original result is still applicable or not, and what happened to change, which all make the study becomes complex. At the same time, as a new interdisciplinary theory, complex network theory is gradually penetrating into numerous fields from biology to sociology [13-15]. Based on this situation, we used sandstone as an example to characterize the

microscopic topological structure of rock seepage network by using complex network theory, and try to study the microcosmic mechanism of macroscopic permeability. Meanwhile, according to the influence of changing external environment on rock pore, we analyzed robustness and its impact on macroscopic permeability.

## 1. The type of rock network structure

The network structure is roughly divided into regular networks and complex networks. Regular network, ER random network and WS small world network are three typical models. We regard $P(k)$ as the probability of a node whose degree is $k$. In the regular networks, the number of connected edges

among nodes is identical, and arbitrary two nodes of it are directly connected. Besides, its average path length is 1, which is very special. In the complex network, each pair of nodes of ER random network is





connected at a given probability $P$, and the vast majority of nodes are $\langle k \rangle$ that is the mean. Most of the nodes of WS small world network are not connected to each other, but different nodes can be reached each other by fewer sides. ER random network and Ws small word network satisfied the Poisson distribution approximatively, however, their average path length and the connection among nodes are

too special. The experimental results have demonstrated that the above homogeneous networks can't effectively explain the widespread real network, such as mobile Internet [16], network of cell metabolism [17-18], Internet networks [19], rock internal structures and so on. BA scale-free network is one of the models of nonuniform networks, very few nodes of it have large degree, and these nodes play a leading role in connectivity of network. Its growth and mechanism of preferential connect are similar to the

characteristics of rock network in real state. Its degree distribution characteristic coincides with the power-law distribution $P(k) \approx k^{-\gamma}$, and there isn't obvious eigenvalue, which leads that it has better fault tolerance rate and robustness to the fault of random node.

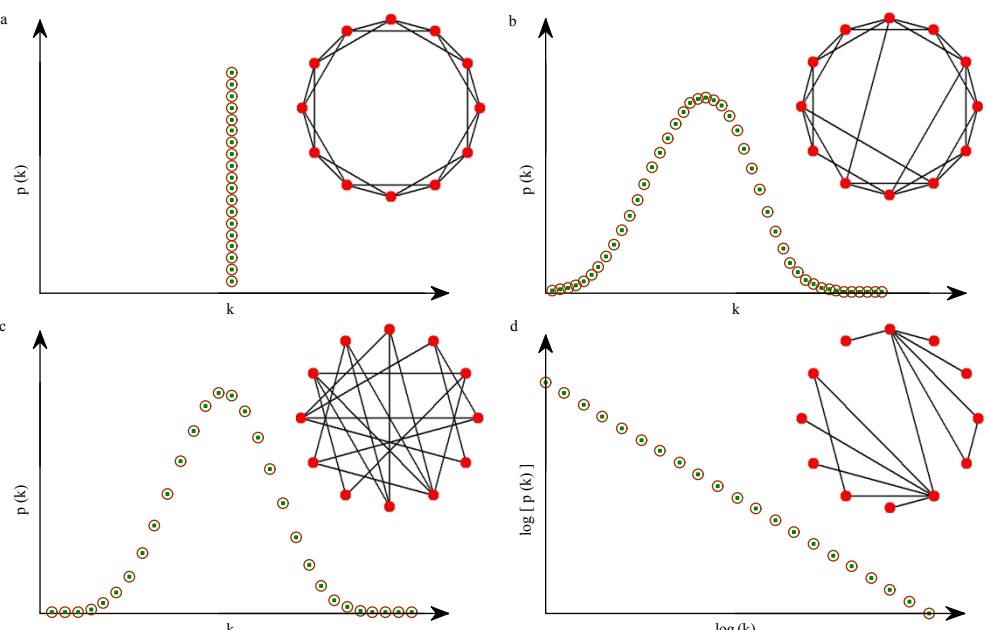

*Fig1:   Network model structure.    (a). regular network[20]. (b). ER random network [21].   (c). WS Small World Network [22] .(d). BA scale-free network [23].*

## 2. Acquirement and simplification of network structure

Imperial College London constructed a binarization model of sandstone pore structure. Image was analyzed by the Micro-CT that produced by Phoenix, Germany, with a 1μm focus system and field of view $512 \times 512$ pixel 8-inch 16-bit detector. Experiments were carried out in nine different porosities of sandstone samples, using different image resolution (from 4.892 to 8.96  μ m) for CT scan, and they obtained nine different the pore structure models with different image accuracy after binarization

processing. For the purpose of testing and comparison, four sandstone models with porosities of 14.1 %,




21.1 %, 24.6 % and 25.1 % were selected in this paper. The following figure demonstrates the three-dimensional sandstone pore network from four kinds of porosities.

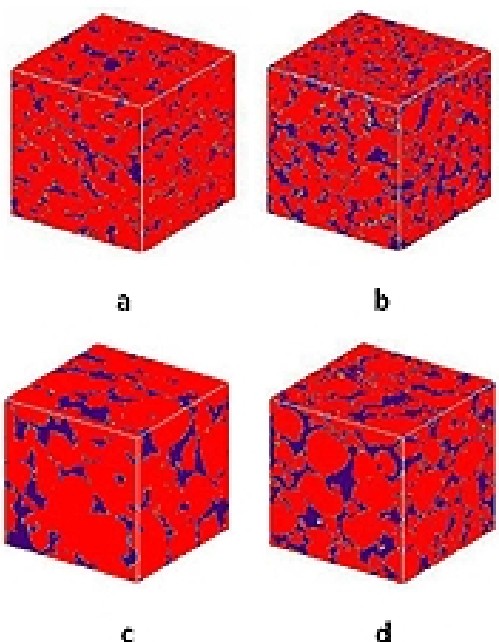

Fig 2: Three - dimensional sandstone pore network. The porosities are (a) 14.1 %, (b) 21.1 %, (c) 24.6 %, (d) 25.1 %.

We used the skeletonization algorithm, regarding pore, throat and the number of pores as node, edge and degree to extract the information of pore structure, and established the model of pore network [24-25]. In the process of studying the network node, we described the degree distribution of the nodes by the distribution function $P(k)$ that means the probability of the degree of an arbitrarily selected node is exactly k. The experiment result demonstrates that the degree of sandstone seepage network of four kinds of porosities satisfies the power-law distribution. It can be seen that the sandstone seepage network belongs to the scale-free network actually ,which means that there may be a small number of hub that play a leading role in the seepage process in the network, and these hub have large degree[26-27] .

In order to simplify the analysis, we removed the edges that connect the node, and we simplified the different edges that connect the two identical nodes into one edge, besides, we removed the nodes whose degree are 1 or 2 as they have no effect on the network seepage. We calculated the process of data simplification such as the sides of seepage network $d$ ,the power exponent $\gamma$ , the mean of nodal degree $\langle k \rangle$, $P(k) \approx k^{-\gamma}$ .The process of data simplification is demonstrated in Fig 3, and the final network statistics that is simplified is demonstrated in Table 1 and Fig 4.



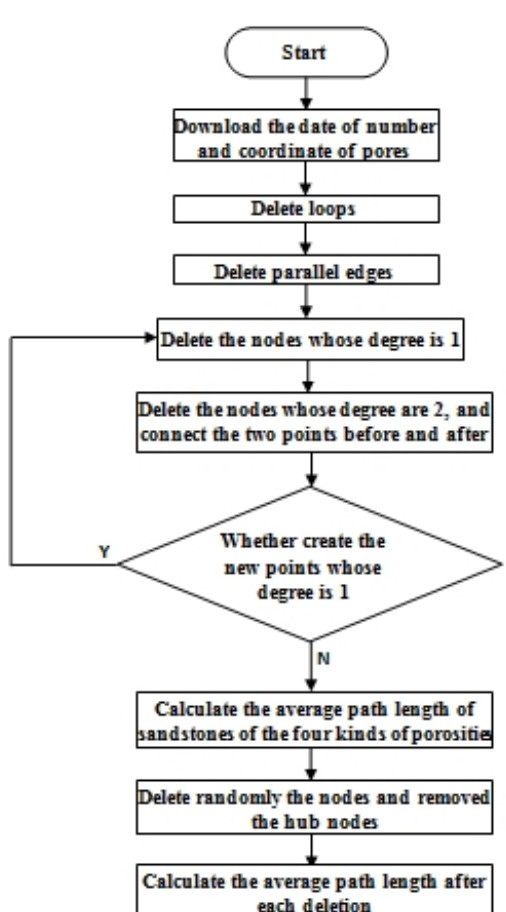

Fig 3. The process of data simplification.

5  Table 1.

| $\phi$ | $d$ | $\gamma$ | $\langle k \rangle$ |
|---|---|---|---|
| 14.1 % | 2241 | 4.7694 | 4.4833 |
| 21.1 % | 674 | 3.2638 | 5.1538 |
| 24.6 % | 4490 | 5.5606 | 5.7786 |
| 25.1 % | 2552 | 3.8674 | 6.1541 |

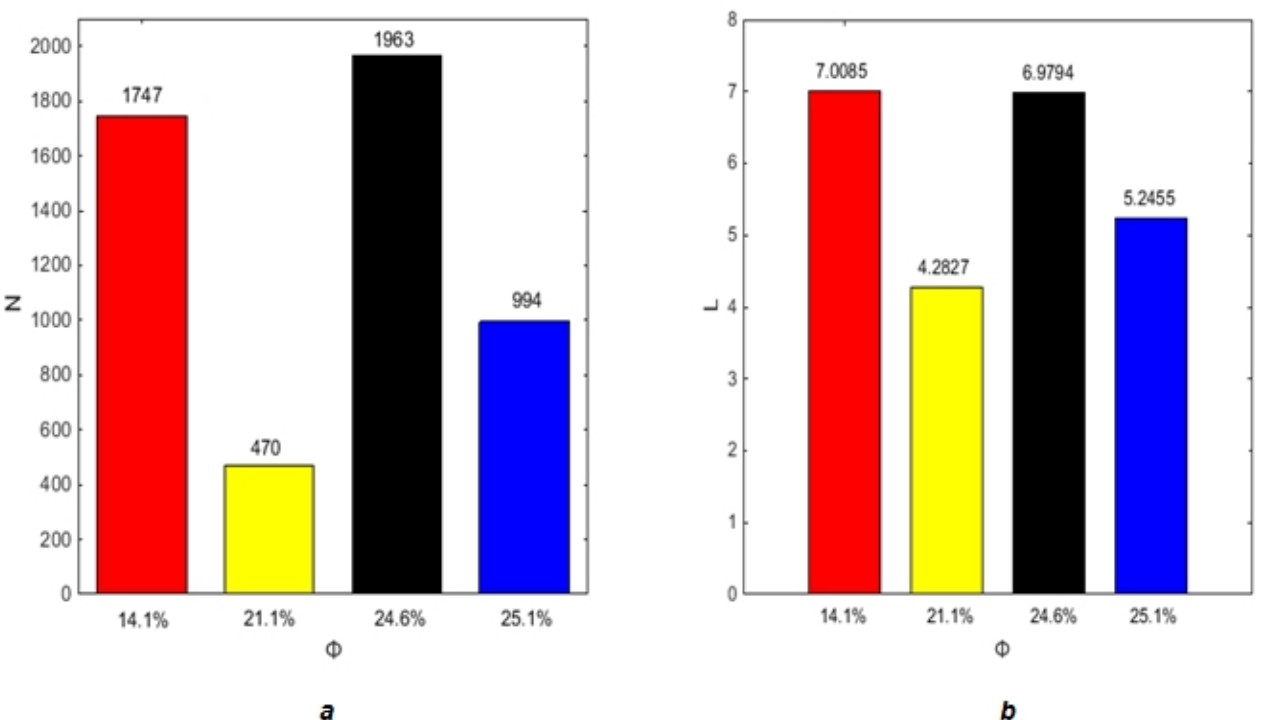

Fig 4: (a). The network magnitude. (b). The average path length $L$ .

## 3. The analysis of the network

When the pores whose connections are complex change, the connectivity of the whole network will changes dramatically. Therefore, the study of the network characteristics such as the distribution of the pore network, the average path length, the shortest edge number among pores and the robustness can help us understand the formation and evolution of the seepage network [28-29].

## 3.1 The analysis about network statistic

The average path length of the sandstone seepage network is defined as the average of the distance between any two pores and it is also the most efficient seepage flow, that is

$$L = \frac{1}{\frac{1}{2}N(N+1)}\sum_{i \geq j} d_{ij}$$

where N stands for the total number of voids in the seepage network, and $d_{ij}$ for the minimum number of halves between the pore $i$ and pore $j$ or the distance between the pore $i$ and pore $j$ .The smaller the average path length of the seepage network,   the less the throat through the connection between the



two pores , which indicates that the smaller the degree of dispersion of the nodes in the network , the easier the fluid seepage.

We used the breadth-first search algorithm to obtain the average path length of the four kinds of
porosities sandstone networks (Stable 1). It can be seen from Fig 4 that the number of sandstone network nodes with porosity of 24.6 % is the maximum and the average path length takes second place in the four kinds of porosities. The number of sandstone network nodes with porosity of 14.1 % takes second place and the average path length takes first place in the four kinds of porosities. The number of sandstone network nodes with porosity of 25.1 % takes third place and the average path length takes
third place in the four kinds of porosities. The number of sandstone network nodes with porosity of 21.1 % takes fourth place and the average path length takes fourth place in the four kinds of porosities. The means of pore degree of the four kinds of porosities sandstones are increasing sequentially. The average length of sandstone generally is proportional to the magnitude of network that is number of nodes. The larger the sandstone network is, the longer the average path length is. Porosity is related to
the network magnitude and the average of node degree, and the average of node degrees plays a dominant role on it. It is presumed that there may be nodes that has a large of degree in the network that is hub that are few in number, but they have a great influence on the average path length among nodes and play a vital role in the seepage process, which also confirms that the sandstone pore network can be describe by the BA scale-free network reasonably. At the same time, the power exponent $\gamma$ is between 3
to 6 whose order is similar to the average path length and the network magnitude, and they haven't change much amount four kinds of porosities (Stable 1 and Fig 5).





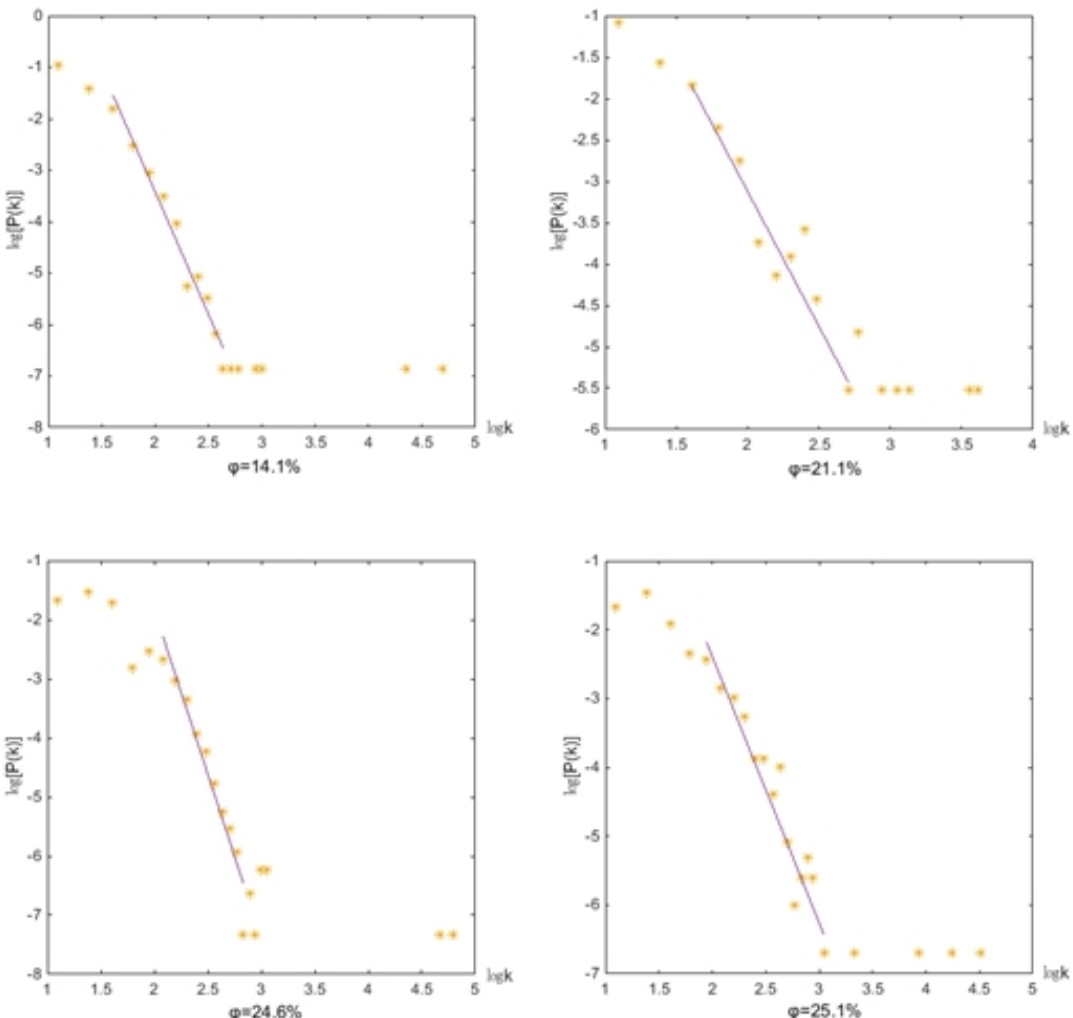

Fig 5: Double logarithm curve of distribution of degree $k$ of different porosities sandstone seepage network.

## 3.2 The analysis of robustness of seepage network under external attack

As the sandstone is affected by natural and human factors in the natural environment, the internal pore network is always changing. While improving the recovery of oil and gas and studying the alteration of pore structure of sandstone under the action of crustal stress need to control the network communication ability within a certain range. Therefore, we studied the robustness that plays an important role in the network communication [30-31]. The attack on the network is fundamentally an attack on the pores, and the targets that are attacked are divided into network hub nodes and non-central nodes [32-33]. In order to simulate the attacks under natural conditions to the maximum extent, we randomly removed the nodes and removed the hub nodes separately for the simplified sandstone pore network. The effect of these two attacks on the average path length is demonstrated in Fig06, the average path length of the seepage


network nodes is increased in general when the nodes are randomly removed, although the increase is very small and even decreases sometimes. And it has little influence on connectivity of the whole network, which indicates that the sandstone seepage network has strong robustness in the face of random attack, and the sandstone seepage networks with different porosities have a similar fault

tolerance mechanism for the reduction of porosity. In contrast, the average path length of the above four kinds of porosities sandstone seepage networks increases rapidly when the nodes with a particularly large degree that is hub are removed. With the remove of nodes, the whole seepage network is decomposed into several relatively isolated sub-networks with similar magnitude. The network connectivity is very different from that of the initial ones, which indicates that the pores with large

degree play a decisive role in the percolation network. It can be inferred that the total seepage capacity of the sandstone pore network does not change greatly when a small amount of pores of sandstone is blocked by the solid particles or the action of ground pressure. While when we increase the number of pores with large degree, even if the number is small, the connectivity of sandstone pores may improve greatly, thereby the seepage rate significantly increase, which has a great reference value for solving the

problems such as the low rate of return in natural resources of oil and gas.



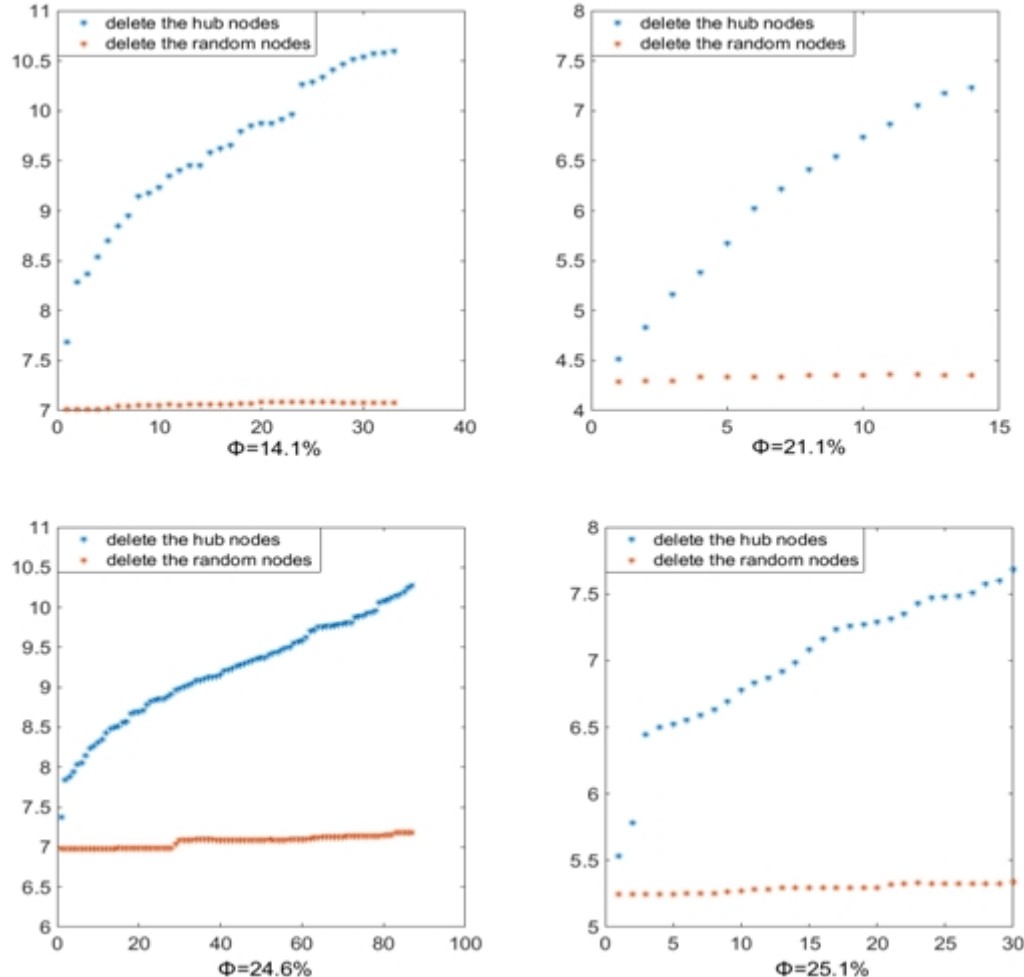

*Fig 6: The alteration of the average path length of the four kinds of porosities sandstones in the random remove of the nodes and the remove of the Hub*

## 5 Conclusions

The topological structure of sandstone seepage network is the result of its physical and chemical properties and external environment. Experiments demonstrated that the topological structure of sandstone seepage network *all* can be described reasonably by the BA scale-free network. The degree of sandstone seepage network with different porosities all satisfies the power-law distribution. The nodes with large degree play a very important role in network connectivity. Sandstone seepage network has strong robustness in the face of external attacks. This conclusion can provide a basis for establishing a reasonable sandstone seepage complex network transport model and using lattice Bertzmann method to simulate sandstone seepage process [34-35]. Although we don't yet know the quantitative relationship between the average path length and the permeability, and there is still a long way to go to accurately





quantify the rock seepage network. However, this paper may provide some reference for the exploration of underground energy mining, pollutant storage and non-rock porous material seepage mechanism from the new perspective of complex network theory.

## Date availability

Our date can be found in the website of Imperial College London:
http://www.imperial.ac.uk/earth-science/research/research-groups/perm/research/pore-scale-modelling/micro-ct-images-and-networks/
We download the standstone S1, standstone S2, standstone S5, standstone S7, and their porosity are 14.1 %, 24.6%, 21.1%, 25.1%.

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

# Acknowledgement

Funding: This work was supported by The Natural Science Foundation of China (NSFC No.11202228 and papd2014).





Table 1.

| $\phi$ | $d$ | $\gamma$ | $\langle k \rangle$ |
| --- | --- | --- | --- |
| 14.1 % | 2241 | 4.7694 | 4.4833 |
| 21.1 % | 674 | 3.2638 | 5.1538 |
| 24.6 % | 4490 | 5.5606 | 5.7786 |
| 25.1 % | 2552 | 3.8674 | 6.1541 |

