# Peer review of "Study on connectivity mechanism and robustness of three - dimensional pore network of sandstone based on complex network theory"

_Nonlinear Processes in Geophysics, 2017_

## Referee Comment (RC1) · Anonymous Referee #1 · 9 Aug 2017

The paper by Liu et al. aims at characterizing pore networks via structural and dynamical measures derived from complex network theory. It is reported that the degree distribution of these networks resembles the ones generated by the BA model; and that the pore networks are robust against random node removals thanks to to its scale-free nature.

The study of spatial networks derived from 3D images is certainly an interesting subject, but I'm afraid the analysis presented in the manuscript is rather poor and, therefore, I do not recommend publication in NPG. In what follows, I explain the points that support this decision:

Despite the countless metrics defined in the complex network literature, the structural characterization in the manuscript is limited to the calculation of the average shortest path length and the degree distribution. Important measures such as transitivity and assortativity are not considered in the study. Furthermore, in my opinion, the random removal of nodes does not add much in the analysis – the percolation dynamics of scale-free networks has been already extensively explored and the fact that pore networks are robust against random failures and highly sensitive to targeted attacks does not come as a surprise.

Throughout the manuscript, one notices some sentences that appear to be fundamental for the analysis but are nonetheless vague. For instance, in page 4 between lines 15 and 20, it is stated:

"In order to simplify the analysis, we removed the edges that connect the node, and we simplified the different edges that connect the two identical nodes into one edge, besides, we removed the nodes whose degree are 1 or 2 as they have no effect on the network seepage. We calculated the process of 20 data simplification such as the sides of seepage network $d$ the power exponent $\gamma$, the mean of nodal degree $\langle k \rangle$, $P(k) \approx k^{-\gamma}$."

Since the goal of the manuscript is to carefully characterize the structure of pore networks, why is this simplification necessary? Perhaps the reason is evident for specialized audience, but anyway I believe this should be clarified for non-experts in the field.

Another odd statement is found in page 7 between lines 19 and 21:

"At the same time, the power exponent $\gamma$ is between 3 to 6 whose order is similar to the average path length and the network magnitude, and they haven't change much amount four kinds of porosities."

I believe it does not make much sense to compare the order of the exponent with

average shortest path length. The same $\gamma$ can yield networks with significantly different scales of $L$. Comparisons should be done between $L$ and the network size $N$, and other metrics such as $\langle k \rangle$, assortativity, etc.

Finally, Figure 5 should have its axes displayed in log-log scale and not in linear scale with the log values; that is, show the results with P(k) and k, not in log P(k) and log k. The quality of the figures should be definitely improved too.

---

## Author Comment (AC1) · 4 Sep 2017

**Study on Topological structure of sandstone 3D pore networks**

Guannan Liu1,2\*, Xiaopeng Pei3, Jishan Liu1,2, Dayu Ye3, Feng Gao1,2, Jianguo Wang1,2

1Mechanics and Civil engineering Institute, China University of Mining and Technology, Xuzhou city, Jiangsu province 221000, China
2 State Key Laboratory for Geomechanics and Deep Underground Engineering, China University of

Mining and Technology, Xuzhou city, Jiangsu province 221000, China

3 Chemical engineering Institute, China University of Mining and Technology, Xuzhou city, Jiangsu province 221000, China

correspondence to Guannan Liu (guannanliu@cumt.edu.cn)

Abstract: There are a large number of pores and throats in rock, the size and shape of which are different, and the connection status is complicated. Based on complex network theory and X ray CT
scanning techniques, we take sandstone as an example to investigate the topological characteristics of sandstone pore networks. The results show that the pore network of sandstone is similar to that of scale-free network. The average path length increases with the increase of network size (number of nodes). It is concluded that a few special pore nodes play an important role in the overall connectivity of the percolation network, and the nodes are ranked by the importance in the network. By analyzing the pore network modularity, it is concluded that the sandstone pore network has a homogeneous

- community structure, which means there is no dominant communities in the networks. The paper try to provide a new perspective for the study of the mechanism of fluid storage and transport in rock and other porous materials.
- 25 Key words: Complex network theory sandstone network topology Seepage distribution of degree

**Introduction:**

10

The pore morphology in rocks are different, cross-scale distributed, and the connection structure is complex[1-3]. It has great theoretical and practical significance to study the characteristics of rock for the accurate and quantitative evaluation of rock permeability, revealing the mechanism of oil and gas accumulation and migration, and enhancing oil recovery[4-6].

In recent years, a lot of research on the statistical characteristics and evolution model of rock pore structure have been done by many scholars. Blunt and Raeini developed a generalized network extraction method for three-dimensional digital core to reduce the uncertainties of modeling[7]. Using a focused ion beam scanning electron microscope (FIB-SEM), Shaina K. [8] obtained a microscale shale digital core and compared it with conventional scanning electron microscopy in terms of porosity, organic content, and pore connectivity. Using statistical method, Hajizadeh took the CT scan image data

40 of sandstone as a sample, and coupled the continuous two-dimensional multi-point statistical simulation

to the multi-scale data extraction program, and proposed a random porous media reconstruction technique[9]. As in the traditional Darcy law is no longer suitable for dense rock, Civan[10] used the modified Darcy law to describe the gas migration in dense shale, in which the apparent permeability is a function of intrinsic permeability and porosity. Considering the basic concept of pore network model, a

- 5 modeling method based on pore volume and throat searching is proposed by Zhang[11], and combined with percolation theory, the permeability of different models is calculated. Yang J.[12] used fractal control function to describe the complex morphology of rock pore structure, and proposed a fractal reconstruction model of rock pore structure with improved simulated annealing algorithm.
- 10 At present, the characterization of pore connectivity structure is mainly studied from macro average perspective or by Eular number. The comprehensive topological characteristics of the porous network such as the degree of clustering and the choice of percolation path are still need to be studied. In this paper, from the point of view of complex network theory, aiming at the real sandstone pore network, the topological characteristics and correlation properties of pore throat network are analyzed to provide the basis for revealing the microscopic mechanism of porous rock seepage.

**1. Type of network structure**

Regular network, ER random network and WS small world network are three classical models to
describe network structure. Regular networks (Fig. 1.a) are equal in degrees for each node, and have minimal average path length (average of the minimum number of edges among all node pairs) in all networks with the same number of nodes . In ER stochastic networks, the degree distribution satisfies the Poisson distribution, and the average path length is larger. As a transition from regular network to stochastic network, the degree of BA small world network (Fig. 1.c) can be approximately represented
by Poisson distribution, especially for the case where the number of nodes is large, and the average path length is very small. As a model for heterogeneous networks, the degree of BA scale-free network

model obeys power-law distribution , and there is no obvious eigenvalue, which can describe the real network topology better. Empirical study [13-16] shows that many real networks are heterogeneous networks.

**5 2. Sandstone pore networks structure**

15

The data of sandstone core constructed by Professor Blunt of Imperial College London is used in our work. Image analysis uses Micro-CT produced by German Phoenix company was used to analysis the image, which equipped with a 1 m focus system with a view of 512 \* 512 pixels with a 8 inch 16 bit

10 detector. Four sandstone samples with porosities of 4.1%, 16.9%, 17.1% and 24.6% are used in our work. The three-dimensional sandstone pore network of the four digital cal is shown in Fig.  $2^{[17-18]}$ .

---

## Referee Comment (RC2) · Anonymous Referee #1 · 18 Sep 2017

The authors have modified the manuscript following some of my suggestions; yet, in my opinion, the significance of the results and their presentation have not improved from the last version. Therefore, I do not recommend the publication.

Here are some points that support this decision:

One of the main conclusions of the paper is that "the sandstone seepage network belongs to a class of scale-free networks". However, by inspecting Fig. 3 one notices that the degree distributions look like single-peaked curves, i.e. the networks are actually more similar to ER and WS networks. Moreover, no statistical test is performed to check whether the degree distributions indeed follow a power-law; the analysis is only

visual and clearly leads to a wrong statement.

The analysis of the eigenvector centrality is also solely performed by means of a visualization. Figure 4 only shows that the distribution of this measurement is different across the networks, but no quantitative indicator is shown. The figure thus ends up being not relevant for the paper.

Still some confusing sentences are found in the text. For instance, in page 7 after line 20: "The closer the connections between adjacent nodes in the network, the larger the density between them, and the relatively independent node modules may form in the process of flow". Adjacent nodes are already connected, so how can they be more closer? And density of what measurement is larger?

Finally, the quality of the figures did not improve from the first version of the manuscript.

---

## Author Comment (AC2) · 25 Sep 2017

Thanks very much for the reviewer's careful work and valuable comments. According to the comments we revised the paper as shown in the attachment.

Please also note the supplement to this comment: https://www.nonlin-processes-geophys-discuss.net/npg-2017-21/npg-2017-21-AC2-supplement.zip

---

## Referee Comment (RC3) · Anonymous Referee #2 · 30 Oct 2017

I have a serious concern about this manuscript as I have the feeling that the authors have a nice idea to study and explore and they haven't analyzed deeply in this subject. The conclusions at the end show that still they have a long run to go. For example, when they say "robustness of seepage network" they don't really analyze it. What they do is a "random attack" and calculate again some network parameters but not a robustness study. I really encourage the authors to improve this initial work to achieve the standards of the journal. Minor comments: Page 1 line 15, you write references [1-3] and normally you don't include references in the abstract. line 19, define "network magnitude" line 23, in this context I suggest to replace the expression "random attacks" for other one more suitable to sandstones. Page 3 line 24, please you should describe

the binarization method. Page 8 Figure 5, how do you select the points to include in the linear regression? which is the error in the estimation of the slopes? Page 10 Figure 6, please include the label in the y-axis

---

## Editor Comment (EC1) · L. Telesca (Editor) · 31 Oct 2017

I don't think the new version of the paper has been significantly improved. The authors did not exhaustively describe the properties of the degree distribution curves, did not test the power-law hypothesis, but simply deleted the section about the description of ER and WS networks, and in the abstract changed the word "scale-free network" with "single-peaked curve". Actually, the problem of testing if such curves follow or not a power-law still exist and the authors did not provide any explanation of it. In Table 1, several measures of complex networks have been reported, but these measures have not been defined qualitatively and quatitatively, and this makes difficult for a reader

who is not familiar with complex netowork theory to follow the contento of the study. The problem of the qualitative analysis of the eigenvector centrality is still present; no quantitative indicator of it has been provided in the revised version. The paper does not in general present a robustness study, as observed by the referee 2. The conclusions are quite weak and do not show how the obtained results could be really useful for the "exploitation of enhancing of the oil and gas recovery". In summary, the paper in its last version does not seem appropriate for the journal.